# The Regulation of the Metastatic Cascade by Physical Activity: A Narrative Review

**DOI:** 10.3390/cancers12010153

**Published:** 2020-01-08

**Authors:** Sophie van Doorslaer de ten Ryen, Louise Deldicque

**Affiliations:** Institute of Neuroscience, Université Catholique de Louvain, Place Pierre de Coubertin 1 L08.10.01, 1348 Louvain-la-Neuve, Belgium; sophie.vandoorslaer@uclouvain.be

**Keywords:** exercise, cancer, tumor, endurance

## Abstract

The purpose of this narrative review is to provide an overview of the currently available knowledge about the mechanisms by which physical activity may affect metastatic development. The search terms exercise [Title/Abstract] AND metastasis [Title/Abstract] returned 222 articles on PUBMED on the 10 February 2019. After careful analysis of the abstracts, a final selection of 24 articles was made. Physical activity regulates the levels of metastatic factors in each of the five steps of the process. Moderate intensity exercise appears to prevent tumor spread around the body, among others, by normalizing angiogenesis, destroying circulating tumor cells, and decreasing endothelial cells permeability. Contrarily, high-intensity exercise seems to favor cancer dissemination, likely through excessive stress, which can be somewhat counteracted by an appropriate warm-up. In conclusion, chronic adaptations to moderate-intensity endurance exercise seem the most effective way to achieve a preventive effect of exercise on metastases. Altogether, the data gathered here reinforce the importance of encouraging cancer patients to perform moderate physical activity several times a week. To limit the undesired events thereof, a good knowledge of the patient’s training level is important to establish an adapted exercise training program.

## 1. Introduction

Cancer is one of the leading causes of death worldwide [1]. Cancer mortality is mainly due to metastases, especially as secondary tumors develop in vital organs. Metastasis is described as the spread of tumor cells, detaching from the primary tumor, circulating through the blood and/or lymphatic vessels, then escaping the circulation to develop a secondary tumor, with the characteristics of the primary tumor, in a distant site within the human body [2]. According to Chambers et al. [3], the metastatic process is composed of five different steps: (1) epithelial-mesenchymal transition; (2) intravasation; (3) surviving in circulation; (4) extravasation, and (5) seeding and colonization. The epithelial-mesenchymal transition and extracellular matrix degradation is the loss of cell-to-cell adhesion (and associated polarity) as well as the acquisition of migratory and invasive properties [4]. Acquiring mesenchymal properties enables the tumor cell to invade surrounding and distant tissues. Conversely, during extravasation, tumor cells undergo a mesenchymal to epithelial transition after the invasion of the distant tissue. Intravasation is the metastatic step where tumor cells enter the systemic circulation via existing or newly formed blood and/or lymphatic vessels to reach and invade a distant tissue [3]. Tumor cells that have reached the lymphatic and/or blood circulation are called circulating tumor cells [5]. Circulating tumor cells acquire some motility and survival characteristics against mechanical and hemodynamic shear stress generated by the blood stream, the immune system, and apoptosis induced by the detachment of tumor cells from the extracellular matrix (also called anoikis), enabling dissemination into secondary tissues [5]. The fourth step of the metastatic process consists in the arrest of the circulating tumor cells that have survived, and their passage through the endothelial membrane to escape the circulation and their subsequent invasion of a distant tissue [6]. Extravasation depends on various factors: mechanical characteristics of blood vessels, such as size and permeability, expression of specific binding factors, and secretion of lytic enzymes, among others. The last step of the metastatic cascade is the seeding of the tumor cell in the distant tissue where it extravasates. The seeding is followed by colonization that needs a favorable microenvironment to proliferate. The ‘seed and soil’ theory [7] proposes that the proliferation of a tumor cell in a secondary tissue is not random and depends on favorable interactions of the tumor cell, which originates from the primary tissue (the seed) and the characteristics of the secondary tissue (the soil). Only an optimal combination of both permits the cell to develop itself and form a secondary tumor. This may explain why certain cancers develop metastases in specific sites and not in others [3].

Following the World Health Organization’s advice, physical inactivity is the predominant avoidable risk factor for developing cancer [8]. In Europe, 9% to 19% of cancers could be prevented if physical activity was performed [9]. A meta-analysis reported that regular physical activity before and after the diagnosis of cancer is associated with reduced cancer mortality, especially in breast-, colon- and endometrial cancer, which depends on the activity level [10]. Furthermore, an increased level of physical activity from before to after the diagnosis, in comparison to subjects who did not change their activity level, was also shown to decrease the risk of cancer mortality [10]. In contrast, a decreased physical activity level between pre- and post-diagnosis was associated with a higher risk of cancer mortality [10]. This altogether reinforces the hypothesis that physical activity, before and after diagnosis, may decrease cancer mortality. 

The links between exercise, cancer risk, and primary tumor metabolism are well studied, but less is known about the regulation of the factors involved in the metastatic cascade by exercise. Therefore, the main purpose of this narrative review is to provide an overview of the currently available knowledge about the mechanisms by which physical activity may affect metastatic development. The secondary aim of this study is to describe the adequate type, volume, frequency, and intensity of exercise to induce beneficial effects on metastatic development.

## 2. Results

### 2.1. Whole Metastatic Development

Three studies looked at the effect of physical activity on the global metastatic development with no focus on a specific metastatic phase. In the first study, a protective effect of physical activity on tumor spread was found in previously trained running mice for 9 weeks, particularly in less aggressive tumor cells [11] (Table 1). In the second study, in a model of breast cancer in mice, 35 weeks-long moderate-intensity exercise training protected from pulmonary metastases, whereas two metastases were measured in the control group [12]. In contrast, in the third study, enhanced tumor growth and metastasis were found in voluntary running mice before (6 weeks) and after (8 weeks) inoculation with liposarcoma cells compared to sedentary mice [13]. Those contrasting results on the effects of physical activity on metastatic development nicely reflect the literature in this research area. The rest of the results section will attempt to present in which conditions beneficial or detrimental effects of physical activity are found for each metastatic phase. Those conditions will then be discussed in the last section.

### 2.2. Epithelial-Mesenchymal Transition and Extracellular Matrix Degradation

Jones et al. investigated whether the observed lower metastatic burden after 8-week voluntary wheel running was related to changes in pro-metastatic gene expression in a model of murine prostate cancer [21]. No clear picture was drawn as a downregulation of the mRNA levels of hepatocyte growth factor receptor, and a tendency towards a downregulation of insulin-like growth factor-1 (IGF-1) receptor, as well as an upregulation of the mRNA levels of C-X-C chemokine receptor type 4 (CXCR4) and a tendency to an upregulation of matrix metalloproteinase (MMP)-9, were measured in the exercising compared to the control group after 8 weeks. Zhang et al. compared the effects of different swimming durations on the levels of dopamine secretion and its effect on liver cancer progression and dissemination in mice [31]. Voluntary swimming (8 min/day for 9 weeks) reduced the rate of metastases and prolonged survival, while forced prolonged swimming (16 or 32 min/day for 9 weeks) had the opposite effect [31]. In the prefrontal cortex, serum, and tumor tissue, the levels of dopamine, known to inhibit angiogenesis, proliferation, and invasion of cancer cells, increased in the voluntary swimming group only. Voluntary swimming decreased the protein expression of transforming growth factor-β1 (TGF-β1), vimentin, and N-cadherin and increased the expression of E-cadherin, while forced prolonged swimming had the opposite effect on the expressions of those molecules in liver cancer tissue. Altogether, those results suggest that extracellular-matrix transition, tumor proliferation, and tissue invasion are slowed down by moderate swimming in mice, while the opposite seems to occur in case of forced prolonged swimming [31].

Extracellular matrix degradation by MMP enhances tumor invasion, tumor cell behavior, and leads to cancer progression. In nasopharyngeal carcinoma cells incubated with platelet-enriched plasma from exercising men, decreased MMP-2 and -9 activities, and increased platelet-nasopharyngeal carcinoma cell aggregation were measured when the subjects had exercised at high-intensity exercise (80%–100% VO_2_max) [33]. Moderate intensity exercise (60% VO_2_max) reduced the formation of platelet-nasopharyngeal carcinoma cells aggregates without changes in MMP activities. Warming-up attenuated the changes induced by high-intensity exercise, possibly by deactivation of adhesion molecules on platelets and regulation of the redox status [33]. 

### 2.3. Intravasation

Angiogenesis is a key process leading to intravasation. Theoretically, the molecular mechanisms regulating angiogenesis may, therefore, influence intravasation, among which hypoxia and the transcription factor responsive to hypoxia, i.e., hypoxia-inducible factor 1 (HIF-1). Higher expression of HIF-1 is associated with an elevated rate of invasiveness and tumor cell motility [6]. Exercise-induced stabilization of HIF-1 is thus expected to increase metastatic spread, but the opposite was observed in voluntary wheel running mice with prostate [21] or breast cancer [20]. The favorable or unfavorable outcome induced by HIF-1 on metastasis seems to depend on the mechanisms by which HIF-1 is activated. Hypoxia induced under resting conditions is linked with pathological and inefficient angiogenesis that sustains a tumor hypoxic microenvironment. In contrast, exercise induces physiological angiogenesis that leads to improved tumor vascularization [34]. Circulating endothelial progenitor cells promote this physiological angiogenesis by incorporating into growing blood vessels [21], but they may also facilitate metastasis by supporting the establishment of the pre-metastatic niche [32]. As the levels of those cells increased after a 12-week aerobic exercise program on a cyclo-ergometer in women suffering from breast cancer, it cannot be excluded that aerobic training could facilitate the acquisition of an invasive tumor phenotype or acceleration of metastasis [32]. 

Chronic physical activity stimulates angiogenesis by increasing vascular endothelial growth factor (VEGF) expression, which leads to a better association between pericytes and endothelial cells and reduced endothelial permeability [17]. In female rats inoculated with mammary cancer, running on a treadmill for 35 weeks induced a higher expression of VEGF-A and higher tumor vascularization, which led to increased tumor growth [17]. However, the tumors were less aggressive than in the sedentary controls, and the time of latency between tumor inoculation and development was longer in exercising rats [17]. Following lung tumor inoculation in mice, 4 weeks of wheel running did not change tumor nor metastasis weight or volume compared to the sedentary group [27]. VEGF is known to upregulate monocyte chemotactic protein-1 (MCP-1) expression, and high MCP-1 plasma levels are associated with lymph node metastases [30]. Voluntary wheel running did not modify the increase in VEGF and MCP-1 after injection of Lewis lung carcinoma cells compared to sedentary mice [30]. 

Another important regulator of intravasation is nitric oxide (NO), which may play a dual and opposite role in this process. On the one hand, NO could promote metastasis by stimulating proliferation, migration, and cellular invasion; on the other hand, the cytotoxic effect of NO on damaged DNA could prevent cancer cell proliferation [35]. A study in mice found that 4 weeks of wheel running after breast cancer cell injection induced a lower NO production together with a higher amount of pulmonary metastases compared to sedentary controls [26]. 

### 2.4. Survival in the Circulation

Regmi et al. looked at the effect of hemodynamic shear stress on the survival of circulating tumor cells, i.e., breast, lung, and ovarian cancer cells, in vitro [16]. Various shear stress conditions were tested, from resting state (15 dynes/cm^2^ in human arteries and 1–6 dynes/cm^2^ in human veins) to intense exercise (up to 60 dynes/cm^2^). Shear stress of 45 and 60 dynes/cm^2^ killed 82% and 99% of the circulating tumor cells, respectively, whereas shear stress of 15 and 30 dynes/cm^2^ allowed 48% and 64% circulating tumor cells viability after the same incubation time in the microfluidic system [16]. Viability was inversely related to the time of incubation [16]. 

Circulating tumor cells are particularly sensitive to immunologic agents circulating in the blood vessels. Thus, exercise could indirectly modulate circulating tumor cells viability via regulation of immunologic factors. In 9-week wheel and treadmill running mice, higher splenic natural killer (NK) cell activity was detected 3 weeks after tumor cell injection and cessation of exercise compared with sedentary controls, which was not associated with reduced tumor incidence in the lung [23]. The effect of NK activity on tumor dissemination seems to be effective only if exercise is initiated before the metastatic spread. Exercise training enhances NK cytotoxicity and levels, but these adaptations do not appear after tumor injection in the blood vessels, possibly due to the suppressive effect of tumor cells on NK activity [36]. 

Macrophages are also important in the control of metastasizing tumor cells. Their cytotoxicity was enhanced in mice after a session of running-to-fatigue at an increasing intensity up to 68%–78% VO_2_max that lasted for about 3 h, and this effect lasted 8 h post-exercise [37]. In the same study, running for 30 min at 55%–65% VO_2_max was less effective at stimulating macrophage cytotoxicity [37]. Similarly, to acute exercise to fatigue, repeated moderate-intensity treadmill running during the 6 days before tumor inoculation enhanced macrophage cytotoxicity and decreased metastatic spread of B16 melanoma cells in mice [25]. 

In sedentary men suffering from nasopharyngeal carcinoma, high-intensity cycling exercise (80%–100% VO_2_max) increased platelet-tumor aggregation and tissue factor-induced coagulation, which are known to promote metastasis [33]. These effects were limited when a warm-up preceded the exercise intervention. Moderate-intensity exercise (60% VO_2_max) inhibited platelet-tumor aggregation and, thereby metastatic dissemination [33]. 

### 2.5. Extravasation

Extravasation begins as the circulating tumor cells get stuck in a blood vessel distant from the primary tumor site. After having found that tumor cell retention in secondary organs post tumor injection was lower in wheel-running trained mice compared to sedentary controls, Hoffmann-Goetz et al. hypothesized that exercise could affect the adherence of tumors to the endothelial vessels [19]. However, others did not find any difference in the localization of metastases between running 30 min at 55%–65% VO_2_max, running to fatigue up to 68%–78% VO_2_max and control mice. A small decrease in the amount of entrapped circulating tumor cells in the run-to-fatigue group was measured [37]. 

Wolff et al. [29] investigated if voluntary wheel-running in mice would modify tight junction protein expression, such as occludin, zonula occludens, and claudin-5, as enhanced tight junctions in blood vessels may prevent the escape of circulating tumor cells into distant tissues [29]. After 5 weeks of voluntary wheel running, mice were injected with highly metastatic tumor cells into the brain vasculature. While a decrease in occludin, zonula occludens, and claudin-5 expression was measured in the sedentary mice, an increase in claudin-5 expression and unchanged occludin and zonula occludens levels were found in the exercising mice 48 h post tumor injection. Three weeks post-injection, expression of occludin and claudin-5 were increased in the exercising mice. As fewer metastases were observed in exercising mice, those results suggest that the protection induced by the blood-brain barrier against metastases could be enhanced. Activation of small GTPases results in claudin-5 and occludin phosphorylation that leads to impaired tight junction proteins function and, thereby decreased endothelial cell tightness and increased blood barrier permeability [38]. In another study in mice, Wolff et al. investigated if 4-week endurance training modulated endothelial permeability after tumor inoculation via the small GTPases Ras, Rac1, and Rho, which are redox-sensitive [28]. Activation of the small GTPase Rho was negatively correlated with running distance in the tumor cells infused mice, probably due to exercise-induced enhanced antioxidant capacity [28]. 

Circulating tumor cells interact with cells such as the bone-resorbing osteoclasts to alter bone remodeling and to invade bones [14]. The effect of mild intensity physical activity on this process has been mimicked in vitro by applying oscillatory fluid flows of 0.8–3 Pa in osteocytes [14]. Conditioned medium from flow-stimulated osteocytes increased migration and reduced apoptosis of breast cancer cells. The opposite was observed with conditioned medium from osteoclasts themselves cultured in flowed osteocytes conditioned medium. Cancer cell trans-endothelial migration was reduced when using flowed osteocytes conditioned medium, which was abolished when blocking intercellular adhesion molecule 1 (ICAM-1) and interleukin 6 in the medium [14]. Those results suggest that osteocyte activation through oscillatory fluid flow can indirectly reduce breast cancer circulating tumor cells migration into the bone matrix and increase circulating tumor cells apoptosis [14]. In a follow-up study looking at molecular regulators of extravasation, the same group found that in addition to ICAM-1, MMP-9 and frizzled 4 were found to regulate invasion of bone-metastatic breast cancer cells [15]. Altogether, flow-stimulated osteocytes can downregulate the bone-metastatic potential of breast cancer cells by signaling through endothelial cells.

### 2.6. Seeding and Colonization

In a recent review, Koelwyn et al. explored the putative effects of exercise in reprogramming the interaction between the host and the tumor microenvironment [39]. Focusing on seeding and colonization, bone marrow-derived dentritic cells seem particularly interesting to study. Those cells regulate angiogenesis, primary tumor, and promote the formation of the premetastatic niche and the colonization of tumor cells. Moderate to intensive cycling training for 12 weeks was shown to increase the amount of circulating bone marrow-derived dentritic cells in women with breast adenocarcinoma, suggesting that exercise could facilitate the acquisition of an invasive tumor phenotype or augment seeding and/or colonization [32]. However, all the aforementioned factors in this result section contribute to the interaction between the host and the tumor environment with more or less importance. As such, a more integrative approach and discussion are needed, which will be the purpose of the next section.

## 3. Discussion

The purpose of this review was to investigate if and how the metastatic cascade is regulated by physical activity with a specific focus on the nature, i.e., aerobic or resistance exercise, and the modalities, i.e., intensity, volume, and frequency, with which exercise is performed (Figure 1). As physical activity was found to either enhance or prevent metastatic spread, it seems that different conditions and modalities of exercise can induce distinctive, and sometimes opposite, effects on the metastatic process, mainly due to specific hormonal responses induced by those different conditions. It is, therefore, crucial to highlight and discuss those specific conditions.

While their levels generally increase after resistance exercise in human [40], IGF-1 and TGF-β1 have been shown to be regulated by voluntary or forced swimming in mice, initiated whether before or after tumor inoculation [31]. Activation of TGF-β1 helps tumor cells to detach from the primary tumor by remodeling the extracellular matrix [31]. IGF-1 promotes metastasis through the inhibition of proteasome-mediated cathepsin degradation. Cathepsin permits tumor invasion and spread by the degradation of the extracellular matrix [41]. However, further investigation is necessary to examine if hormones secreted following resistance exercise may directly enhance extracellular matrix degradation in the tumor microenvironment, certainly in human. Catecholamines secreted during intensive exercise can regulate inflammatory cytokines, such as TNF-α and interleukins (i.e., IL-1 and IL-6) [42]. These hormones enhance endothelial permeability and adhesion to the vascular wall, helping circulating tumor cells to enter in a distant tissue. Increased secretion of catecholamines may also result in the enhancement of NK cell activity and cytotoxicity, as NK cells express β-adrenergic receptors [43]. Voluntary running suppresses tumor growth and spread through epinephrine- and IL-6-dependent NK cell mobilization and redistribution, which modulates immunity and increases efficiency against circulating tumor cells [43]. Higher NK cell activity has been linked with a lower rate of metastasis [11]. Finally, inflammatory cytokines and growth hormones may also increase MMP expression, which plays a role, among others, in extracellular matrix degradation [44]. Their expression was shown to be either increased, either decreased in physically active patients, but directly correlated with TNF-α levels. The following sections will discuss how different exercise modalities may regulate the metastatic cascade.

### 3.1. Moderate Versus High Intensity Physical Activity

In healthy people, higher intensity leads to higher levels of exercise-induced stress and, thereby more adaptations and a more potent increased in exercise capacity. However, in cancer patients, exercising at higher intensities does not always appear to be beneficial when it comes to tumor spread capacities. Exercise intensity may affect platelet adhesion to tumor cells as the aggregation of platelets has been found to be higher than resting levels after high-intensity endurance exercise [33]. When the intensity was moderate, platelet adhesion was lower, thereby decreasing circulating tumor cells’ survival in the vessels. Of note, platelet-tumor aggregation was decreased below resting levels after high-intensity exercise if the subjects performed a warm-up before [33]. 

In addition to platelet adhesion, exercise intensity also modulates immunity [18,22,24], which is especially important in the survival of circulating tumor cells. A higher phagocytic capacity of macrophages was reported after moderate-intensity exercise, but not after high-intensity treadmill running nor in sedentary controls [45]. It is important to note that the macrophage phenotype may either enhance (M2 macrophages) or prevent (M1 macrophages) tumor growth and dissemination. Both moderate- and high-intensity exercise decrease macrophage recruitment in the tumor microenvironment and enhance anti-inflammatory M1 phenotype expression and macrophage cytotoxicity [46].

The regulation of adhesion molecules, such as ICAM-1, is dependent on exercise intensity as well. ICAM-1 plays a role in endothelial barrier disruption, cell adhesion, and transmigration [47] and thus, promotes extravasation. In breast cancer cells, ICAM-1 activation was lower after moderate-intensity electrical stimulation in vitro [14,15]. In cancer patients with an activity level of 150 or 300 min/week at 50%–70% of maximum heart rate (i.e., moderate-intensity exercise), ICAM-1 levels were lower, which in turn reduced the amount of circulating tumor cells in the vessels [48]. A recent review concluded that moderate-intensity endurance exercise in healthy people is associated with a decrease in adhesion molecules, whereas high-intensity endurance exercise is associated with an increased expression of these factors for several hours post-exercise [47]. In addition, interval training is more effective at reducing ICAM-1 levels than continuous training. Whether those conclusions apply to cancer patients needs to be tested as well as the possible involvement in the modulation of the metastatic development.

### 3.2. Acute Versus Chronic Physical Activity

The responses to physical activity can be categorized into short-term adaptations, during or directly after a single exercise session, and long-term adaptations, when exercise sessions are repeated over a certain period. One typical example of this differential adaptation according to the repetitive character or not is the immune response. Circulatory immune markers have been shown to decrease directly after an exercise session, but the immune response is improved after a training period [49]. In addition to the chronicity of physical activity and the training status, other factors may affect the immune response to acute exercise, including age, nutritional status, and extreme environments, but the most critical determinants are the intensity and duration of the exercise bout [49]. The intensity mainly affects changes in blood lymphocyte numbers during and after exercise, while the duration has a stronger influence on the neutrophil and total leukocyte count [50]. It is, therefore, important not to exercise at a too high intensity or during a too long period to avoid a significant drop in the immune activity as the latter would favor cancer cell survival in the circulation and metastatic spread. The difficulty is to determine the intensity and the duration threshold for each patient, which mainly depends on the training status and the progression of the disease.

Regular moderate exercise is usually associated with reductions in circulating pro-inflammatory cytokines, increased T-cell proliferation, and NK cell activity, leading to enhanced immuno-surveillance in resting state [49]. Indeed, immunity is a first response to eliminate abnormal cells that develop into a malignant tumor mass and tumor cells [51]. Furthermore, immune cells mobilized by exercise tend to be more differentiated and to have an increased cytotoxic function. In mice, cytotoxic immune cells are particularly mobilized after wheel running by increasing the number of immune-attractive chemokines and ligands that activate NK cells [43]. 

In addition to the immune system, chronic exercise has been shown to regulate the Wnt-β-catenin pathway. The latter regulates the primary step in the metastatic process, i.e., tumor invasion, and, more particularly, the regulation of cell differentiation, polarity, and cell-to-cell adhesion [47]. After wheel-running for 6 weeks, levels of E-cadherin were higher, and nuclear levels of β-catenin were lower in small intestine tumors in mice [52]. Together with the higher expression of the tumor suppressor E-cadherin [52], the lower expression of β-catenin indicates that tumor invasion is reduced after 6 weeks of wheel running in mice. In human, low levels of nuclear β-catenin in normal and colorectal cancer cells of exercising patients were associated with reduced mortality risk [53]. These results confirm previous findings that patients with low nuclear β-catenin levels and high physical activity levels (>18 MET-h/week) after cancer diagnosis had lower cancer mortality rates [54]. This all suggests that chronic exercise may be beneficial for controlling tumor invasion by decreasing β-catenin levels. Here as well, both the chronicity and the intensity of physical activity seem important in the regulation of tumor invasion.

Vascularization, an important long-term adaptation in response to exercise training, has been shown to play a predominant role in the second step of the metastatic cascade. Angiogenesis can be induced by NO, which usually increases in response to both acute and chronic physical activity [17]. In aerobic exercising breast cancer patients, higher NO production was related to less aggressive and invasive tumors [32]. Inversely, lower NO production after training was associated with higher metastatic rates in mice [26]. The latter result is in line with the inhibitory role of NO in platelet aggregation, together with its vasodilatory and anti-oxidative properties [26]. High NO production would reduce while low NO production would rather favor metastasis development. While those results seem consistent regarding metastasis, it remains to determine why exercise training for a few weeks induces opposite results on NO production, which is probably not only a matter of model, i.e., patients vs. mice. Other unknown mechanisms are probably involved.

### 3.3. Forced versus Voluntary Physical Activity

Forced physical activity relates to treadmill running or to any structured exercise, while voluntary activity refers to wheel running or to any exercise spontaneously performed. Forced prolonged physical activity may exert deleterious effects through excessive physiological and psychological stress and consequently counteract the positive effects of exercise [38]. For example, voluntary swimming for 8 min was linked with decreased metastatic burden, whereas forced prolonged swimming time enhanced tumor growth and lung metastatic spread in mice with transplanted liver cancer [31]. After voluntary exercise, levels of dopamine, which exhibits anti-tumor properties, were increased, and levels of TGF-β1, a key factor to induce epithelial-mesenchymal transition, decreased [31]. Those levels evolved in the opposite direction in the forced swimming group [31]. No resting time was allowed for the whole duration of forced prolonged swimming sessions compared to voluntary swimming, which usually allows intermittent floating and recovery. Sufficient recovery time between exercise sessions also needs to be taken into account to avoid excessive fatigue and potential deleterious outcomes in cancer patients [55].

Based on the previous sections, it seems particularly important to adapt the training program to the training status of the patient and to increase progressively the volume, the intensity, and the frequency of the training sessions as well as the recovery between the sessions.

### 3.4. Lactate Levels

When dealing with cancer and physical activity, it is impossible not to deal with lactate as the latter plays key roles in both conditions. Contrary to what was thought for years, lactate is not a waste product of exercise but rather an important energy substrate for gluconeogenesis, which can be oxidized in type I muscle fibers as well [56]. In addition, the buffer capacity of lactate limits the pH decrease during exercise [56]. In cancer, lactate plays a major role in tumor proliferation and malignant phenotyping, i.e., self-sufficient metabolism, angiogenesis, immune escape, migration, and metastasis [57]. Its modulation by exercise may thus have an impact on tumor invasion. The link between exercise-induced lactate metabolism and tumor spread has been reviewed by San-Millan and Brooks [57]. For more details, the reader is referred to this review, but the key message is that lactate is an important factor of tumor proliferation and dissemination and, therefore, a new main target for cancer therapies. Its role in cancer metabolism is comparable with exercise and forms a new field of investigation in exercise-oncology but needs to be better understood. Blood lactate levels are directly correlated with exercise intensity. It can thus be hypothesized that, on the one hand, moderate-intensity exercise may improve tumor lactate metabolism and mitochondrial function; on the other hand, high-intensity exercise may lead to overexpression of lactate receptors, thus enhancing its action in tumor growth and spread. Based on the latter and the current knowledge, it is probably safe not to recommend high-intensity training to cancer patients.

### 3.5. Limitations

A first limitation of the study was the heterogeneity in the investigated variables that together make it difficult to compare the different studies. Tumor types, method, and timing of tumor inoculation, investigated species, exercise protocols, among others, were found to differ across studies. These differences can also help to explain divergent results among studies investigating similar factors. A second limitation is the fact that most studies were conducted in vitro or in rodents. Observations may, therefore, not entirely reflect the same processes in humans subjected to the same kind of protocols. Third, no study evaluated the effect of resistance training alone on metastatic development while the latter is regularly included in the exercise training programs for cancer patients.

### 3.6. Perspectives

Regarding resistance exercise, it is known that it activates the mammalian target of rapamycin (mTOR) pathway to increase protein synthesis and, in the long term, muscle mass [58]. As the mTOR pathway is also highly active in proliferating cancer cells [59], it will be crucial to determine whether the activation of the mTOR pathway by resistance exercise does not exacerbate tumor proliferation and metastasis development. Another perspective is the better understanding of the role of myokines in metastasis development. Some myokines, such as decorin, IL-6, irisin, oncostatin-M, have been found to play a role in cancer modulation [59,60]. Due to their systemic effects, those myokines form a new interesting research area in the regulation of tumor cell metabolism. Finally, knowing that exercise-induced beneficial effects on tumor spread may vary according to the type of tumor and the step in the metastatic process [30], further investigation will be required to adjust the exercise prescription accordingly.

## 4. Materials and Methods 

The search terms exercise [Title/Abstract] AND metastasis [Title/Abstract] returned 222 articles on PUBMED on the 10 February 2019. A first selection was made based on title relevance and language. Articles not written in English were excluded. Sixty-two articles remained after this first selection round. This was followed by a careful reading of the abstracts, which reduced the number of articles from 62 to 31. The final selection was made by reading the 31 remaining articles, and 24 original articles were finally included (Table 1). Studies were selected for their relevance in terms of adaptations to exercise and metastatic development. Studies with a focus on primary tumor growth and those with mixed conditions, e.g., diet and exercise combined with no exercise group alone, were excluded. Finally, two studies were excluded because they only described the protocol of the experiment, and no result was presented. Applying the same searching methodology in Embase and Scopus resulted in no additional article.

## 5. Conclusions

This narrative review investigated if acute and chronic physiological changes in response to physical activity regulate metastatic spread. Chronic adaptations to moderate-intensity endurance exercise (60%–70% VO_2_max) seems the most effective way to limit excessive stress and to achieve a preventive effect of exercise on metastases, whereas high-intensity exercise (>60%–70% VO_2_max) was shown to enhance the metastatic spread in some cases. Altogether, the data gathered here reinforce the importance of encouraging cancer patients to perform some form of moderate physical activity several times a week. To limit the undesired events thereof, a good knowledge of the patient’s training level is important to establish an adapted and progressive exercise training program, with sufficient recovery between exercise sessions. 

## Figures and Tables

**Figure 1 cancers-12-00153-f001:**
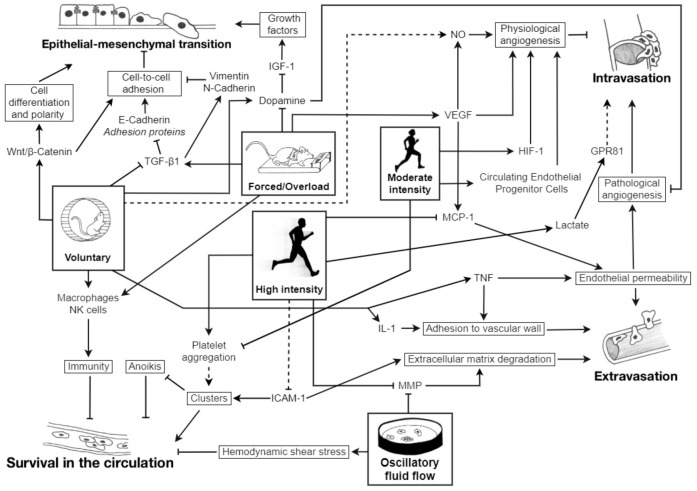
Schematic overview of the regulation of the metastatic cascade (epithelial-mesenchymal transition, intravasation, survival in the circulation, and extravasation) by different exercise modalities in vitro, in rodent and in human. ICAM-1: intercellular adhesion molecule 1; GPR81: G-protein coupled receptor 81; HIF-1: hypoxia inducible factor-1; IGF-1: insulin-like growth factor-1; IL: interleukin; MCP-1: monocyte chemotactic protein-1; MMP: matrix metalloproteinases; NK: natural killer; NO: nitric oxide; TGF-β1: transforming growth factor β1; TNF: tumor necrosis factor; VEGF: vascular endothelial growth factor; Wnt: Wingless and Int. →: activation; 
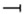
: inhibition; plain lines ―: demonstrated; striped lines - - -: hypothetical.

**Table 1 cancers-12-00153-t001:** Design and main results from the studies selected based on the research equation.

Reference	Model	Tumor Type	Exercise Intervention	Measured Effects	Main Results (vs. Sedentary)	Conclusions
	In vitro					
Ma et al. (2018) [14]	In vitro	Breast cancer cells (non-metastatic and highly metastatic)	In vitro oscillatory fluid flow 1 Pa (mimicking mechanistic loading during mild exercise)	Osteocytes activity Osteoclasts activity IL-6 and ICAM-1	Mimicked loading ↗ osteocytes activation Osteocytes directly ↗ cancer cell migration and ↘ apoptosis Osteocytes inhibit osteoclasts differentiation → ↘ migration and ↗ apoptosis Ostéocytes ↘ IL-6 expression → ↘ ICAM-1 → ↘ apoptosis	Opposite effects on breast cancer cell migration and apoptosis directly and indirectly mediated by osteocytes exposed to oscillatory fluid flow
Ma et al. (2018) [15]	In vitro	Bone metastatic breast cancer cells and osteocytes	Oscillatory Fluid Flow (2 h at 1 Hz and 1 peak shear stress at 1 Pa)	Endothelial permeability Adhesion on endothelial monolayer Cancer cells genes expression	↘ Endothelial permeability ↘ Breast cancer cell adhesion onto endothelial monolayer (mediated by ICAM-1) Alteration of cancer cell genes expression via endothelial cells ↘ MMP-9 and FZD4	Flow-stimulated osteocytes ↘ bone-metastatic potential of breast cancer cells by signaling through endothelial cells
Regmi et al. (2017) [16]	In vitro	Breast cancer (+ lung metastases) Ovarian lung cancer cells Leukemic cancer cells	Microfluidic circulatory system: Low shear stress of 15 dynes/cm^2^ (resting state) High shear stress of 60 dynes/cm^2^ (intensive ex)	CTC necrosis CTC apoptosis	↗ CTC death in high shear stress 90% necrosis within 4h with high shear stress 10% apoptosis within 16–24h ↘ Viability of highly metastatic tumor cells in prolonged high shear stress treatment	Intensive exercise may be a good strategy for generating high shear stress that can destroy CTC and prevent cancer metastasis
	**Rodents**					
Alvarado et al. (2017) [12]	Female rats	Mammary cancer cells, ER and PR positive tumor cells	35 weeks of moderate exercise training (treadmill running), 60 min/day, 5 days/week; post-tumor injection	Primary tumor and metastases development ER and PR immunoexpression	↘ Mammary tumors numbers and masses No metastasis developed in exercising animals vs. 2 developed in the sedentary group ↗ ER and PR immuno-expression in neoplasms from sedentary and exercising groups	Long-term exercise training ↘ the risk of metastatic dissemination of breast cancer Anti-metastatic effects of exercise training are hormone-independent
Assi et al. (2017) [13]	Male mice	Liposarcoma	6 weeks of spontaneous physical activity before tumor injection 8 weeks of voluntary wheel running post-tumor injection	Intramuscular tumor size FABP4 IL-6, C/EBP-β PPAR-γ Autophagy markers	Larger intramuscular tumors ↗ Expression of FAPB4, C/EBP-β, and PPAR-γ ↗ IL-6 levels in both active and inactive groups ↗ Expression of autophagy markers Beclin-1 and GABARAPL-1	Physical activity ↗ liposarcoma development by ↗ autophagy in tumor mass
Faustino-Rocha et al. (2016) [17]	Female rats	Mammary cancer	35 weeks of treadmill running (60 min/day, 5 days/week)	VEGF-A Vascularization Aggressiveness	↗ VEGF-A expression ↗ Tumor vascularization ↘ Aggressiveness	Long-term exercise training: ↗ Tumor vascularization ↘ Tumor multiplicity and aggressiveness
Hoffman-Goetz et al. (1994) [18]	Female mice	Mammary tumor line 66 (in vivo) YAC-1 (in vitro)	8 weeks before tumor inoculation: forced treadmill (T), voluntary wheel running (W) or sedentary (S) 3 weeks post-inoculation: continuation (TT/WW), cessation (TS/WS), start activity (ST/SW) or maintenance sedentary (SS)	NK cell activity LAK Pulmonary tumor density	Exercise before tumor injection: ↘ Basal NK activity (WS, TS) ↗ LAK activity (WS, TS) ↘ Pulmonary tumor density Exercise continued after tumor injection: ↘ LAK activity (TT, WW) ↗ Pulmonary tumor density (TT, WW)	↗ LAK activity with endurance training or physical activity before tumor injection ↗ Natural immunity with exercise training but no significant difference in tumor burden and spread Exercise-induced changes depend on the tumor sensitivity to NK and LAK cells
Hoffmann-Goetz et al. (1994) [19]	Male mice	CIRAS 1 tumor cells	9 weeks of voluntary wheel running before tumor cells injection	Tumor cell lung residency Tumor cell radioactivity	↘ Lung tumor cell residency in exercise-trained animals at 5 min post-injection, up to 30 min ↘ Tumor cell radioactivity in liver, spleen, kidney of wheel running mice 30 min and 3 h post-injection	Exercise training ↘ tumor cell adherence to the vascular wall
Jadeski et al. (1996) [11]	Mice	CIRAS 1 and CIRAS 3 Beige mutation (impaired NK)	9 weeks of treadmill running (30 min/day, 5 days/week) before tumor cell injection	Tumor cell retention in lungs NK cell contribution to exercise-mediated effects	↘ Tumor retention in trained (mutated or not) mice ↘ Tumor cell retention in running CIRAS 1 mice vs. sedentary No effect of training in CIRAS 3 retention	Exercise training ↗ innate natural immune responses Physical activity is protective early in the metastatic cascade, before selection of a more aggressive and metastatic phenotype
Jones et al. (2010) [20]	Female mice	Human mammary adenocarcinoma	Voluntary wheel running	Tumor volume Tumor vascularization Hypoxia Tumor energy status	No difference in tumor volume ↗ tumor vascularization Higher intratumoral hypoxia levels (HIF-1) No difference in ATP, PGC-1α and AMPK levels	Aerobic exercise ↗ intratumoral vascularization → normalization of tumor microenvironment → ↘ rate of metastasis and ↗ cancer therapy efficiency
Jones et al. (2012) [21]	Male mice	Murine prostate cancer	8 weeks of voluntary wheel running after tumor inoculation	Tumor blood perfusion HIF-1 and angiogenesis Prometastatic mRNA levels Tumor MAPK and PI3K signaling Circulating proinflammatory cytokines + metabolites	↘ Expression of pro-metastatic genes in exercising animals Improved tumor vascularization associated with ↗ intratumoral levels of HIF-1α and VEGF ↗ Expression of metabolic genes in tumors ↘ Plasma angiogenic cytokines	Exercise-induced stabilization of HIF-1α upregulates VEGF expression. This led to physiological tumor vascularization with a shift toward suppressed metastasis.
MacNeil et al. (1993) [22]	Mice	CIRAS 1 tumor cells with pulmonary metastases	Voluntary wheel running (W) or sedentary (S) 9 weeks pre- and 3 weeks post tumor inoculation four groups: WW, WS, SW, and SS	Lung tumor number Natural immunity (NK cell activity, splenic serine esterase)	↘ Number of lung metastases in WW and WS No difference in splenic esterase activity, nor in NK cell activity	Exercise training before, but not after tumor inoculation ↘ number of lung metastases
MacNeil et al. (1993) [23]	Male mice and in vitro	CIRAS 1 tumor cells YAC-1 tumor cells	9 weeks before tumor injection - Continuous access to wheel - Treadmill exercise, 30 min/day, 5 days/week No exercise for 3 weeks post-tumor injection	Lung metastasis Lung tumor cell retention NK cell spleen number and cytotoxicity	No change in lung metastases incidence ↗ Number of tumors with running distance Small ↘ tumor cell retention in lung ↗ Splenic NK cells cytotoxicity but no change in number	Exercise ↗ the development of metastases Low clinical impact of enhanced immunity on tumor growth and spread
MacNeil et al. (1993) [24]	Male mice	CIRAS 3 tumor cells YAC-1 tumor cells	9 weeks before tumor injection, 30 min/day, 5 days/week - Voluntary wheel running - Forced treadmill running	Citrate synthase activity Tumor cell retention In vivo and in vitro NK cell cytotoxicity	Citrate synthase activity: treadmill running > wheel running = sedentary ↘ Lung tumor cell retention ↗ In vitro splenic NK cytotoxicity	Exercise training ↗ natural immunity and NK cell cytotoxicity against tumor cells and ↘ pulmonary tumor retention in wheel and treadmill running mice
Murphy et al. (2004) [25]	Male mice	B16 melanoma cells	6 days of treadmill running (1 h/day) before tumor cells injection	Metastatic spread Macrophages cytotoxicity	↘ Amount of lung metastases ↗ Macrophages cytotoxicity	Short-term moderate-intensity exercise training ↘ metastatic spread by ↗ macrophages function
Smeda et al. (2017) [26]	Female mice	Orthotopic breast cancer cells	4 weeks of voluntary wheel running after cancer injection	Tumor volume and number NO production Platelet activation	↗ Number in pulmonary metastases associated with ↘ NO production in aorta No significant ↘ in systemic NO bioavailability No change in plasma P-selection concentration and platelet activation markers	Pro-metastatic effect of voluntary exercise associated with lower NO production Potential explanations: - ↗ ROS and RNS production - Pseudo-vessels formation - Untrained animals
Tsai et al. (2013) [27]	Male mice	Lewis lung carcinoma	4 weeks aerobic training, 5×/week, post-inoculation - Interval: 6 × 10 min at 65% maximum speed - Continuous: 45 min at 54%–58% VO_2_max	VEGF Tumor growth and metastasis	↗ Serum VEGF levels in both exercising groups but not in tumor VEGF levels No difference between exercising conditions No difference in weight and volume of lung and liver tumors	No effect of moderate exercise on tumor growth despite higher plasma VEGF levels
Wolff et al. (2014) [28]	Male mice	Lewis lung carcinoma	4 weeks of voluntary wheel running before tumor inoculation	Oxidative status of brain microvessels Anti-oxidative enzymes gene expression Rho-GTP-ase activation	↘ Superoxide levels in high running group ↘ Antioxidant capacity in tumor cell infusion + exercise ↘ Levels of Rho-GTPases activation in high running group	Exercise can protect microvessels from blood-brain barrier instability by decreasing Rho activation
Wolff et al. (2015) [29]	Male mice	Murine Lewis lung carcinoma	5 weeks of voluntary wheel running before tumor inoculation	Tight Junction proteins - Occludin - ZO-1 - Claudin-5	48 h post tumor cells injection: - Maintained expression of occludin and ZO-1 in exercised tumor mice - ↗ Claudin-5 expression with exercise 3 weeks post tumor cells injection: - ↗ Basal levels of occludin and claudin-5	↗ Blood-barrier regulation by exercise ↘ Extravasating tumor cells in the brains from exercised mice
Yan et al. (2011) [30]	Male mice	Lewis lung carcinoma	9 weeks of non-motorized voluntary running before tumor inoculation 2 weeks of voluntary running after tumor inoculation	Plasma angiogenic cytokines (PDGF, VEGF, and MCP-1) Insulin, leptin, adiponectin	No differences in number and size of metastasis ↘ Plasma insulin and leptin levels ↗ Adiponectin levels ↗ PDGF but not VEGF and MCP-1	No effect of voluntary running on metastasis Voluntary running may affect favorably energy expenditure and adipogenesis (against obesity)
Zhang et al. (2016) [31]	Mice and in vitro	Liver cancer transplantation with lung metastasis	9 weeks of regular moderate (8 min/day) and overload (16 and 32 min/day) swimming Swimming before and after cancer inoculation	Lung metastases ratio DR2 activation TGF-β1 expression	↘ Mean lung metastases ratio in 8 min-group and ↗ in 16 and 32 min-groups DR2 activation ↘ cancer cell proliferation and invasion Moderate swimming ↘ EMT induced by TGF-β1	Divergent activation of dopamine system may explain the opposite effects on tumor growth and metastasis between moderate and overload swimming
	**Human**					
Jones et al. (2013) [32]	Women	Breast adenocarcinoma	▪ 20–45 min cycle ergometer, at 55% to 100% of VO_2_max, 3x/week, 12 weeks	Tumor blood flow Tumor hypoxia CEP Plasma cytokines and angiogenic factors IL-1β, IL-2, PLGF	↘ Tumor hypoxia and enhanced tumor vascularization ↗ CEP, PLGF, ↘ IL-1β, IL-2 No change in tumor phenotype markers (CD-31, HIF-1, GRP78) ↘ Gene expression of NF-κB signaling and inflammation	Aerobic exercise training can modulate tumor progression and metastasis: - ↗ Provascular adaptations - Normalized tumor microenvironment
Wang et al. (2007) [33]	Men	Nasopharyngeal carcinoma (NPC)	Bicycle ergometer - Moderate-intensity exercise (60% VO_2_max for 40min) - High-intensity exercise (up to 100% VO_2_max) - Warm-up exercise (20 min at 60% VO_2_max → 30 min rest → VO_2_max)	Platelet-NPC aggregation Platelet-promoted Tissue Factor TFPI MMP-2 and -9 Tissue Inhibitor of MMP-1	High-intensity exercise: - ↗ Binding affinity to fibrinogen - ↗ Tissue Factor expression/activity - ↗ Tissue Inhibitor of MMP release - ↘ TFPI release - ↘ MMP-2 and -9 activities Warm-up ↘ these effects Moderate-intensity exercise: ↘ formation of platelet-NPC aggregates	High-intensity exercise: - ↗ Aggregation and coagulation - ↘ MMP bioactivity ↘ Effect of HIE after warm-up Moderate-intensity exercise: Minimizes the risk of thrombosis induced by platelet-NPC interactions

Characteristics of tumor cell types: CIRAS 1: sensitive to NK-cell lysis, less aggressive than CIRAS 3; Mammary tumor line 66: highly metastatic, NK-cell resistant. C/EBP-β: CCAAT-enhancer-binding proteins; CEP: circulating endothelial progenitor cells; CTC: circulating tumor cells; DR: dopamine receptor; EMT: epithelial-mesenchymal transition; ER: estrogen receptor; FABP4: fatty acid-binding protein 4; FZD4: frizzled-4; HIE: high-intensity exercise; HIF-1: hypoxia-inducible factor 1; ICAM-1: intercellular adhesion molecule 1; IL: interleukin; LAK: lymphokine-activated killer; MCP-1: monocyte chemotactic protein-1; MMP: matrix metalloproteinases; NK: natural killer; NO: nitric oxide; NPC: nasopharyngeal carcinoma; PDGF: platelet-derived growth factor; PPAR-γ: peroxisome proliferator-activated receptor coactivator gamma; PR: progesterone receptor; RNS: reactive nitrogen species; ROS: reactive oxygen species; TFPI: tissue factor pathway inhibitor; TGF-β1: transforming growth factor β-1; VEGF: vascular endothelial growth factor; ZO: zonula occludens.

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
