# Peer review of "The Regulation of the Metastatic Cascade by Physical Activity: A Narrative Review"

_cancers, 2020, doi:10.3390/cancers12010153_

Round 1

Reviewer 1 Report

In the manuscript entitled "The regulation of the metastatic cascade by physical activity: a narrative review", the auothers summarized the studies regarding how physical acitivity affects metastatic behavior of tumors. The authors systematically searched previous studies and selected the most relevant results for discussion. They introduced how physical activities affect the five steps of metastasis including epithelial-mesenchymal transition, intravasation, surviving in circulation, extravasation and, seeding and colonization. Overall, this review manuscript comprehensively summarized previous and current research results. I have a few minor concerns: 

1) As the authors showed moderate intensity exercise is beneficial to patients, but high intensity exercise is detrimental and favorable for metastasis. What is the standard for the strength of excercise so that patients could know how to excercise? 

2) Almost all the 24 references used animal models for study, can you also use other search engines to get more references to see if there are more studies and results which are based on human so that it will be more convince and clinically relevant?

Author Response

Reviewer 1

In the manuscript entitled "The regulation of the metastatic cascade by physical activity: a narrative review", the auothers summarized the studies regarding how physical acitivity affects metastatic behavior of tumors. The authors systematically searched previous studies and selected the most relevant results for discussion. They introduced how physical activities affect the five steps of metastasis including epithelial-mesenchymal transition, intravasation, surviving in circulation, extravasation and, seeding and colonization. Overall, this review manuscript comprehensively summarized previous and current research results. I have a few minor concerns: 

As the authors showed moderate intensity exercise is beneficial to patients, but high intensity exercise is detrimental and favorable for metastasis. What is the standard for the strength of excercise so that patients could know how to excercise? 

Several studies in animals (13, 23, 26, 31) showed enhanced metastatic spread following a single exercise or exercise training. The conditions that enhanced metastatic spread were forced wheel running or swimming for prolonged duration. These conditions led to excessive stress, which really seems to be the key that triggers exercise-induced harmful instead of beneficial effects. If subjects progressively increase the intensity of exercise training within and between exercise sessions, stress will be minimized.

A nice example of this comes from a study in women with breast adenocarcinoma, in which exercise intensity was progressively increased up to 100% VO2max (19). During the first week, 3 sessions were realized at 55-60% VOâ‚‚max. Then, the intensity of one or two sessions per week increased to 65-75% VOâ‚‚max, to reach in the four last weeks, one session a week at 100% VOâ‚‚max. Regarding tumor metabolism, they observed that the primary tumors were less aggressive but also better vascularized than in the non-exercising group. While this study did not specifically investigate metastatic spread, it suggests that progressive increase in training load up to max is feasible in patients without increasing tumor aggressivity, the latter being known to promote metastasis spread.

In addition, warm-up and cool-down may play a role in reducing stress induced by exercise at high intensities. In a study in which subjects performed an exercise of moderate or high intensity (16), platelet aggregation and circulating tumor cell survival were shown to be decreased in the “moderate intensity” group and in the “warm-up + high intensity” exercise group, in contrast to the subjects who did not warm-up before high intensity exercise.

Recommendable intensity for endurance exercise depends on the training status of the patient. An untrained subject should begin with moderate intensity exercise, around 60-70% VOâ‚‚max. Sufficient recovery time between exercise sessions is also important, especially in cancer patients.

In conclusion, exercising at high-intensity (>70-80% VO2max) may have opposite effects on the metastatic spread, depending on the ability of the subject to counteract exercise-induced stress. Progressive training, warm-up and recovery time will help to reduce this stress.

We have now specified the intensity in the conclusion of the manuscript as well as the importance of reducing stress and allowing sufficient recovery between the sessions: “This narrative review investigated if acute and chronic physiological changes in response to physical activity regulate metastatic spread. Chronic adaptations to moderate intensity endurance exercise (60-70% VO2max) seems the most effective way to limit excessive stress and to achieve a preventive effect of exercise on metastases, whereas high intensity exercise (>60-70% VO2max) was shown to enhance the metastatic spread in some cases. Altogether, the data gathered here reinforce the importance of encouraging cancer patients to perform some form of moderate physical activity several times a week. To limit the undesired events thereof, a good knowledge of the patient’s training level is important to establish an adapted and progressive exercise training program, with sufficient recovery between exercise sessions. “

Almost all the 24 references used animal models for study, can you also use other search engines to get more references to see if there are more studies and results which are based on human so that it will be more convince and clinically relevant?

The search terms “Exercise (title, abstract, author keywords) and Metastasis (title, abstract, author keywords)” returned 165 reviews and articles on Embase, and 213 on Scopus. Selection based on the same criteria as those used for PubMed gave no additional article. This has been added to the section methodology “Applying the same searching methodology in Embase and Scopus resulted in no additional article".

Reviewer 2 Report

This review article entitled “The regulation of the metastatic cascade by physical activity: a narrative review” by Sophie van Doorslaer de ten Ryen and Louise Deldicque is a comprehensive review about an association between physical activity and the development of metastasis. The descriptions on this paper are correctly based on the previous papers. This paper is well-organized and will be of interest to readers of cancers. I have only a few concerns mentioned below.

Comments

1. The article is well-organized and written in details, however, the present form includes a mix of the results of in vitro studies and the patient data. Explicitly stating them helps readers understand the content more easily and precisely. For example,  adding such information to Table 1 is an option to be considered. 

2.   The cancer type and its hormone sensitivity may affect the metastatic cascade because physical activity triggers the secretion of a variety of hormones. Summary of the association between the hormone secretion by exercise and the cancer metastasis would be useful for readers. 

Author Response

Reviewer 2

This review article entitled “The regulation of the metastatic cascade by physical activity: a narrative review” by Sophie van Doorslaer de ten Ryen and Louise Deldicque is a comprehensive review about an association between physical activity and the development of metastasis. The descriptions on this paper are correctly based on the previous papers. This paper is well-organized and will be of interest to readers of cancers. I have only a few concerns mentioned below.

Comments

The article is well-organized and written in details, however, the present form includes a mix of the results of in vitro studies and the patient data. Explicitly stating them helps readers understand the content more easily and precisely. For example,  adding such information to Table 1 is an option to be considered. 

Table 1 has been reorganized following the suggestion: at first, in vitro studies, then studies made in rodents and finally, in human.

The cancer type and its hormone sensitivity may affect the metastatic cascade because physical activity triggers the secretion of a variety of hormones. Summary of the association between the hormone secretion by exercise and the cancer metastasis would be useful for readers. 

A paragraph summarizing principal associations between hormone secretion by exercise and the role of these hormones in the metastatic cascade has been added at the begin of the discussion:

“While their levels generally increase after resistance exercise in human [41], IGF-1 and TGF-β1 have been shown to be regulated by voluntary or forced swimming in mice, initiated whether before or after tumor inoculation [31]. Activation of TGF-β1 helps tumor cells to detach from the primary tumor by remodeling the extracellular matrix [31]. IGF-1 promotes metastasis through the inhibition of proteasome-mediated cathepsin degradation. Cathepsin permits tumor invasion and spread by the degradation of the extracellular matrix [42]. However, further investigation is necessary to examine if hormones secreted following resistance exercise may directly enhance extracellular matrix degradation in the tumor microenvironment, certainly in human. Catecholamines secreted during intensive exercise can regulate inflammatory cytokines such as TNF-a and interleukins (i.e. IL-1 and IL-6) [43]. These hormones enhance endothelial permeability and adhesion to vascular wall, helping circulating tumor cells to enter in a distant tissue. Increased secretion of catecholamines may also result into enhancement of NK cell activity and cytotoxicity, as NK cells express β-adrenergic receptors [44]. Voluntary running suppresses tumor growth and spread through epinephrine- and IL-6-dependent NK cell mobilization and redistribution, which modulates immunity and increases efficiency against circulating tumor cells [44]. Higher NK cell activity has been linked with lower rate of metastasis [11]. Finally, inflammatory cytokines and growth hormones may also increase MMP expression, which play a role, amongst others, in extracellular matrix degradation [45]. Their expression was shown to be either increased, either decreased in physically active patients, but directly correlated with TNF-a levels. The following sections will discuss how different exercise modalities may regulate the metastatic cascade.”

Reviewer 3 Report

This manuscript though straightforward is cursory at this time due to lack of any specific relevance. The authors should avoid overstatement of the results and while inferring need to consider that correlation does not necessarily mean causation.

The authors haven’t presented the manuscript in an intelligible fashion. The manuscript lacks logical presentation of ideas. This makes it difficult to understand the intent of authors’ message.

According to this reviewer, the manuscript in its current form doesn’t deserve to be published.

Author Response

We regret that the manuscript did not respond to the expectations of the reviewer. Based on the comments above, we understand that it would be impossible to satisfy the reviewer. As no specific comment was made, no modification could be brought to the manuscript.

Round 2

Reviewer 1 Report

Thanks for the clarification and improvements of the manuscript.

Reviewer 2 Report

This revised version of the review article entitled “The regulation of the metastatic cascade by physical activity: a narrative review” by Sophie van Doorslaer de ten Ryen and Louise Deldicque is a comprehensive review about an association between physical activity and the development of metastasis. The descriptions on this paper are correctly based on the previous papers. 

The authors revised the manuscript completely according to the reviewers’ comments. The revised version of the manuscript is well-written and of scientific value. This report would be of interest to readers of cancers.